# Historical Evolution of Cattle Management and Herd Health of Dairy Farms in OECD Countries

**DOI:** 10.3390/vetsci9030125

**Published:** 2022-03-09

**Authors:** Ivo Medeiros, Aitor Fernandez-Novo, Susana Astiz, João Simões

**Affiliations:** 1Veterinary and Animal Research Centre (CECAV), Department of Veterinary Sciences, School of Agricultural and Veterinary Sciences, University of Trás-os-Montes and Alto Douro (UTAD), 5000-801 Vila Real, Portugal; al62766@utad.eu; 2Department of Veterinary Medicine, School of Biomedical and Health Sciences, Universidad Europea de Madrid, C/Tajo s/n, Villaviciosa De Odón, 28670 Madrid, Spain; aitor.fernandez@universidadeuropea.es; 3Animal Reproduction Department, National Institute of Agronomic Research (INIA), Puerta De Hierro Avenue s/n, CP, 28040 Madrid, Spain; astiz.susana@inia.es

**Keywords:** dairy cattle, animal production, health management, production systems

## Abstract

This work aimed to review the important aspects of the dairy industry evolution at herd level, interrelating production with health management systems. Since the beginning of the industrialization of the dairy cattle sector (1950s), driven by the need to feed the rapidly growing urban areas, this industry has experienced several improvements, evolving in management and technology. These changes have been felt above all in the terms of milking, rearing, nutrition, reproductive management, and design of facilities. Shortage of labor, emphasis on increasing farm efficiency, and quality of life of the farmers were the driving factors for these changes. To achieve it, in many areas of the world, pasture production has been abandoned, moving to indoor production, which allows for greater nutritional and reproductive control of the animals. To keep pace with this paradigm in milk production, animal health management has also been improved. Prevention and biosecurity have become essential to control and prevent pathologies that cause great economic losses. As such, veterinary herd health management programs were created, allowing the management of health of the herd as a whole, through the common work of veterinarians and farmers. These programs address the farms holistically, from breeding to nutrition, from prevention to consultancy. In addition, farmers are now faced with a consumer more concerned on animal production, valuing certified products that respect animal health and welfare, as well as environmental sustainability.

## 1. Introduction

There are currently around 265 million dairy cows worldwide, of different breeds, producing approximately 906 million tons of milk in 2020 (latest available data), almost in comparison to the 50-year-old production levels [1]. By the second half of 20th century in US, 21% more animals and 23% more feed were required to produce a billion kilograms of milk, when compared to current production regimes and techniques [2]. As a result of management and production methods, we have more efficient animals, presenting higher individual production, and more efficient systems, which have led to competitiveness and economic robustness of the dairy farms. There are fewer, but larger, farms than ever before, reducing fixed production costs [3], and, consequently, increasing incomes.

Several changes, including the introduction of new practices and technologies, were implemented, which allowed the increase in productivity per animal and per farm. These changes have been observed in the milking routine, feeding practices, breeding, housing conceptions, and health management programs, among others. In the large milk production centers (e.g., European and American continents, and China) there has been a complete shift from traditional pasture-based production systems, towards indoor production systems [4,5]. These allowed for a rapid specialization and mechanization of the milk production process. In these systems, there is a greater possibility of implementing veterinary herd health management (VHHM) programs, including reproduction, prevention, and nutrition management, and veterinary consultancy. However, in areas such as Ireland [6], New Zealand [7], or even the Azores [8], due to their climatic and geographic characteristics, pasture-based systems are still the most common production systems. However, these do not allow for such effective control of the health and performance indicators of the animals, and are frequently linked to a delay in the implementation of new management practices and technologies [9].

All of these changes and improvements in the production systems came as a response to the needs of producers, who struggled to increase their yields, due to the low price paid per liter of milk [3]. Furthermore, it came as an alternative for the lack of labor seen in the livestock sector worldwide, as this sector is heading toward specialization and mechanization in the future [10,11].

This paper aims to review the historical evolution of the bovine dairy sector up to the present day, focusing on changes in the production systems (pasture vs. indoor), reproduction, nutrition, genetics, calves, and heifers rearing management, and adult dairy cow management, and their interrelations with herd health and production management, thus highlighting disease prevention and biosecurity measures. The first part of this review focuses on the current global dairy sector emphasizing OECD countries, which historically have been pioneers in herd health approach, including the EU, where the specific EU quotas system was abolished in 2015. The interrelationships between the different aspects of animal production and VHHM are discussed in the second part.

## 2. The Current Worldwide Dairy Sector

### 2.1. Dairy Sector Dynamics and Milk Consumption

The economic and social importance of the dairy sector is significant. According to the American Dairy Association North East [12], in 2021, this sector was responsible for employing around 900,000 people in the US. However, this strength had progressively decreased within one century. In 1900, 41% of the US workforce was employed in agriculture, but in 2000, only 2% were employed in this primary sector [13]. This was due to the increasing mechanization of the sector, reaching mass production records [14]. In the EU, the dairy sector represents approximately 12% of the output coming from agriculture [15].

In 2020, 906 million tons of cows’ milk [1] were produced worldwide, with an expected growth of about 1.6% per year in the current decade [16]. Asia is the leading producer, producing about 42% (379 million tons) of the total milk yield, including even buffalo milk (*Bubalus bubalis*) can reach half of the milk yield in some countries, such as India (90 million tons, according to Faostat; Figure 1). Europe produces approximately 26% (236 million tons), followed by North America with 12% (111 million tons; US, 101 million tons), and South America with 9% (82 million tons) of total milk yield [1]. However, despite the socio-economic impact caused by the COVID-19 pandemic, in 2020 there was an increase in the international dairy trade of milk and its derivates (1.2% more). This was mainly due to the growth in demand in the Chinese market, based on the improved average quality of life, expanded consumer base, higher consumption per capita within that population, and an increased demand of milk power used in piggeries [1].

Although milk has been consumed historically in rural areas by farmers, only the development of the pasteurization process by Louis Pasteur in 1864 [17] made it possible for the milk supply and its derivates to be consumed by the urban population from the beginning of the 20th century [18]. Pasteurization turned milk into a safe product, extending its lifetime and preventing zoonotic diseases [19].

Milk is a highly energetic nourishment which is rich in nutrients, and has been part of humanity’s diet for hundreds of years [20]. Milk consumption during adulthood, in lactose-tolerant humans, brings immense health benefits, both in bone strengthening and in the prevention of cardiovascular diseases, cancer, diabetes, and others [21]. Despite this, milk consumption in some parts of Africa, the Middle East, and Asia is reduced, due to cultural reasons, i.e., dairy products are not traditionally included in diet, socio-economic reasons [22], and also because in these parts of the world, there are large populations without lactase persistency, thus causing lactose intolerance [23]. Nevertheless, according to Faostat [24], in the last decade (2010–2019), the Asian food supply quantity of milk and dairy derivate products (excluding butter) progressively increased from 37.3 to 45.9 kg/per capita/year. Lactose-free or lactose-reduced has been one of the solutions from the dairy industry to feed lactose intolerant people [19,25].

### 2.2. EU Milk Quotas, Their Abolition, and the Economy of Scale

In 1984, the EU implemented the milk quota system (i.e., limitation of milk yield production or supply to dairy) to control the rising milk production at the time, ensuring a minimum milk price in periods of high production, regardless of market demand, came about [26]. Following the end of milk quotas in the EU 2015, the price of milk has undergone moments of great uncertainty and volatility [27].

From 1983 to 2013 there was a reduction in approximately 81% (1.2 million fewer farms) of dairy farms in the EU [15]. The abolition of quotas has triggered a global alignment of milk prices [28]. These events, and the existence of an increasingly competitive world market, have triggered several changes in the dairy industry [29]. This has motivated farmers to produce more efficiently, change management and routines, and identify failures and weakness [30].

Therefore, farmers have increased herd size and the average yield per cow per year [3]. Enlarging herds is the most common strategy to increase gross income, as well as maintain economical sustainability. In the future, these increasing herd sizes will be notable in low income per farm countries, such as India and Pakistan, where it is expected that they will represent 30% of world production by 2029 [31].

The worldwide intensification in the dairy sector in recent decades has also brought negative consequences, such as local environmental pollution, and constrained ecological sustainability [32]. This is the opposite of what consumers look for. Consumers, besides an adequate price, would like to have safe products from farms complying with animal welfare, being both environmentally sustainable, and eco-friendly [15,33].

## 3. Milk Production: From Rearing to Milking

### 3.1. Dairy Cattle Breeds

Since the creation of dairy farms, the sector has worked on and improved several breeds, in order to be able to produce huge amounts of milk. Holstein/Holstein–Friesian, Jersey, Brown Swiss, Guernsey, and Ayrshire are the main dairy breeds worldwide [5]. Jersey cows are specialized in milk with a high content of solids (more 0.57% of total protein and, above all, more 1.37% of total fat, when compared to Holstein–Friesians), and show high reproductive performance [34]. However, out of all the breeds, McClearn et al. [35] showed that Holstein–Friesian cows are the most efficient and profitable (5720 vs. 5476 kg per cow of total milk yield, from Holstein–Friesian vs. Jersey × Holstein–Friesian, respectively) breed for mass milk production, when compared to crossbreeds with dairy aptitude.

Due to these traits, the Holstein is the dairy breed of choice for worldwide farmers [36]. The origin of the Holstein breed dates back 2000 years ago, in The Netherlands, where Germanic tribes needed to obtain animals capable of making the best value of their lands [37]. Therefore, they crossed black animals, “Batavians”, with white animals, “Friesians”, obtaining more efficient animals with greater productive capacities [38]. In 1852, the first specimens of this breed arrived in Boston. It was in USA that the main genetic improvement took place, transforming the Holstein cows into that what we currently have [38,39]. Diverse reproductive biotechnologies, such as artificial insemination (AI) and embryo transfer, have allowed the genetic improvement of the breed, as well as its dissemination all over the world [36].

### 3.2. Genetic Improvement of Dairy Cattle

All of the improvements made at the genetic level of dairy breeds enabled farms to improve their production levels [40]. These improvements began to be realized at the beginning of the 20th century, through the selection of animals with higher production traits, and with a higher content in milk solids [41]. This trend was observed throughout the 20th century, thus the genetic choice fell, mainly in productive traits and conformational traits [41,42]. An example of this is that, in 1920, Holstein cows produced on average 2000 kg of milk annually (305 days in milk). In comparison, 100 years later, Holstein cows on average produce over 10,000 kg of milk annually, with the same content of solids [41]. Through artificial insemination techniques, it was possible to disseminate and cross bulls of superior genetic value all over the world [42]. This cross-breeding resulted in genetic improvement through the production of in genetically superior offspring [43].

This fast genetic improvement, mainly focused on increasing milk yield in the Holstein cattle [32], has led to associated problems, such as poor fertility, inadequate health, and shorter longevity. As a result, from the beginning of the 21st century onwards, genetic improvement underwent a radical transformation, with a greater focus on reproductive traits (fertility), health (average SCC), and sustainability traits (alimentary efficiency) [40].

The genetic evaluation of the sires can be performed in a traditional way, evaluating the animal’s pedigree and progeny testing [44]. Progeny testing is based on the performance of the offspring, which, when superior to the parental generation, leads to genetic improvement [42,44]. Recent technologies have made it possible to evaluate genetic merit or value through genomic prediction [45], whereby, through the analysis of the genome of the animals, the predicted breeding values (EBV) can be estimated [44]. Due to the high amount of data available, as well as pedigree information, and phenotypes information, the estimation of the breeding values of the selected animals is highly reliable [46]. In addition to being a better tool for the selection of the desired traits and breed conservation, it restricts inbreeding, which negatively impacts reproduction and production traits [47].

Furthermore, producers started, in some parts of the world, to crossbreed [48]. Crossbreeding occurs when two different lines, breeds, or populations are crossed [49]. This technique brings some benefits, such as introducing greater genetic variability of desired characteristics, especially when there is a high inbreeding depression [48]. Crossbred dairy cattle have been considered more robust (10 to 15% of expected heterosis for longevity in Danish models) and more fertile (about 10% of expected heterosis), as well as economically efficient, when compared to full breeds [49,50,51]. According to Dezetter et al. [52], Montbéliarde and Normandé have been the most frequently used breeds in the EU to cross with Holstein cows. These breeds, originating from France, show higher resilience, better fertility, produce milk with more solids, and have dual purpose, providing an additional income for farmers when these animals are culled and sold for beef [52]. Nevertheless, high-yielding cows have also been selected for many years, for other genetic traits, related to health, longevity, reproduction, and other positive characters besides yield capacity [43,53], and can also show high fertility rates [54]. Moreover, improving nutritional, welfare, and reproductive management appears to be the key to sustain fertility in dairy farms [55]. Moreover, new methodologies (e.g., genetic selection of best adapted animals) are being currently tested to improve breed health and resilience and reduce the footprint of production [32,56,57].

As such, a new focus is on more sustainable and balanced production, with a reduction in the ecological footprint of farms [58]. Thus, in response to the current needs of industries and consumers, genetic selection has begun to give greater emphasis to animal welfare, health, longevity, resilience, and environmental efficiency (methane emission and food efficiency) traits [58].

### 3.3. Rearing Calves and Heifers

Similar to the evolutionary process, the process of rearing has experienced several substantial changes in the last decades, in terms of housing and feeding calves and heifers based, again, on increased knowledge concerning the physiology, pathology, and requirements of the animals at each life stage [59]. Although rearing is the most sensitive sector of a dairy herd [60], on many farms, the poor management of calves leads to high mortality and morbidity rates [61].

Heifer rearing represents, on average, 15 to 20% of the total costs of a farm, bringing no immediate economic benefit, as heifers do not produce milk until the first calving [62]. A recent study (2020) calculated the cost of rearing a heifer from birth to first calving, with automatic feeders, but in different housing systems. It concluded that rearing costs are higher in confinement (USD 1920 on average) than in pasture (USD 1335 on average) [63]. In The Netherlands (2015), a similar cost was estimated to be around EUR 1790 [64]. Higher costs (2017) were reported in the UK (GBP 1819 ± 387; up to less than GBP 407.83 in herd sizes ≥100 milking cow) [65].

Nonetheless, these animals are subjected to a very high risk of morbidity and mortality, due to poor management concerning the rearing [66]. Neo-natal diarrhea and bovine respiratory disease (BRD), are the two main diseases that affect calves [67,68,69]. These conditions are complex and of multifactorial origin, which makes it control difficult for both farmers and veterinarians [69]. To reduce these incidences, several management protocols are advised [70,71]. The passive transfer of immunity from cow to calf, through colostrum, is an example of one of the key points of these improved management protocols.

Colostrum is the first milk produced by the dam, and this is characterized by being rich in proteins and antibodies [72]. The ingestion of this highly important and rich nourishment during the first 6 h of life [60] reduces the risk of infection for many pathologies [73]. However, colostrum supply is not a simple process, and it must be rigorous, since poor colostrum management is frequent, and leads to failure in the transfer of immunity [73]. Good and clean colostrum management is essential for the survival and vitality of calves [72]. These good practices are characterized by providing high quality colostrum (rich in immunoglobulin G (IgG), i.e., a minimum of 50 mg/mL IgG), in an adequate volume [74], and provided within the optimal range for IgG absorption during the first 6 h and up to 24 h after birth. Beyond 24 h of life, the IgG absorption is very limited [72], and, when possible, colostrum should be pasteurized [75].

Furthermore, current guidelines advise that, from the first week of age, calves should be provided with *ad libitum* clean water, starter concentrate, good quality straw or hay, and milk supply three times a day, with a total of 7 to 10% of body weight [76,77,78]. *Ad libitum* clean water supply is fundamental immediately after the birth of the calves, as it stimulates the ingestion of dry food or concentrate, which is vital for rumen development, rapid growth, and vitality [77]. A recent study [78] concluded that calves reared with the practices mentioned presented bigger hip height, larger body length, and better digestion. These animals continued to present better growth rate and better health five months later [78]. Recent works have evaluated the supplementation of fatty acids in milk and concentrate [79], or the use of cobalt-lactate [80], with the objective of reducing the use of antibiotics, and increase the health status of calves. These studies showed that the animals had greater efficiency, better health, and an increased growth rate.

In addition, improvement in calves’ housing and implementation of preventive protocols based on vaccines have been developed [67,73]. The most common type of calf housing in the EU and in the USA is the individual housing of the new-born calf based, on observed good health results and weight gain [81,82]. This is a biosecurity measure, since this type of housing reduces the transmission of respiratory and gastrointestinal diseases, albeit limiting the animal’s natural behavior [61]. A recent study compared the risk of contamination by diarrheal viral agents, and there was a lower risk in individual housing when compared to group housing [83]. Despite this, individual housing is considered as restraint for calves’ welfare. Recent studies have demonstrated that group rearing is better for calves’ social and behavioral development [84]. Pairing housing systems, being the current recommended practice for rearing, improves calves’ health and welfare [82,85]. These systems increase positive social behavior, promoting the intake of solid feed, higher average daily gain, and reducing fearfulness of animals [86,87]. Rearing calves even in groups from a very young age is now being considered, since this practice allows for less labor, distress, and the adoption of new feeding technologies [88]. One important technical advancement with rearing has been the automated milk feeders (AMF) [59,67]. These automated systems make it possible to increase the frequency with which calves are milk fed, increase animal welfare, and reduce the workload of producers [89]. The increase in milk intake by calves allows them to grow at a faster rate, and allows for a reduction in the incidence of diseases such as diarrhea [28].

Vaccination programs have been carried out to increase the passive immunity of calves against neo-natal diarrhea via the vaccination of dams in the prepartum period [60]. Vaccinating calves against the agents causing bovine respiratory complex, or directly in calves during the first three weeks of age with intranasal vaccines [68], is another common suggestion. Thus, all of these advances in the knowledge about rearing management have allowed farmers to improve their work, increased animal welfare, and reduced the incidence of the two main diseases that affect cattle at these young ages. The calves and heifers are the future of the farm, and those not affected by diarrhea or respiratory diseases early on become healthier and more productive adult cows, bringing more income to farmers [90].

### 3.4. Pasture-Based Systems versus Indoor Systems

Since the first dairy farms came into production, the sector has undergone an intense transformation. In the beginning, farms were pasture-based, with grassland being the main constituent of cattle feed [5]. Currently, this system is still implemented in regions of great preponderance in the dairy sector, and in available pastures, such as New Zealand [7], Ireland [6], Uruguay [91], and the Azores [8] (Figure 2). These regions allow the cultivation of annual rich pastures [5,8], due to the mild climate and abundant rainfall throughout the year.

In dry and very hot regions, where the soils are poor, such as the west coast of the USA, Mexico, south of Europe, and the Middle East, dairy farms are permanently indoors, with production based on stored forage and external cereal systems [5].

Thus, we observe two main production systems, depending on the main source of feeding: the traditional pasture-based system, and the indoor-system with indoor feeding.

In general, pasture-based farms report better animal welfare indicators [92,93,94,95], showing lower incidence of pathologies, such as lameness, hoof injuries, uterine diseases, and lower mortality rates, than the rates from indoor farms [96]. Cows are grazing animals, with pastures being their natural environment, where they can express their physiological behavior, reducing distress and immunological depression [97]. This system is also known for being low-cost, benefiting from natural resources [98].

However, systems exclusively using pasture have limitations as well. Cows fed exclusively or mostly grazing may not meet their nutritional needs, thus reducing their yield [99], and the supplementation of their diets is quite complicated in grasslands. This reinforces the need to supply feed, so that they can cover their nutritional needs for health and yield [100]. Another limitation of pasture-based systems is the dependence of grass growth on climate conditions, making it much more difficult to provide a homogenous daily feed (Figure 3) [101]. The implementation of mechanized systems is still more difficult and less common in grazing farms [102]. All of this makes the pasture-based systems low output systems, as compared to the indoor systems [98].

The thermal stress that animals are subjected to outdoors causes discomfort, decreased performance, and decreased milk production [103]. To assess the thermal comfort of the cows, a temperature–humidity index (THI) was established, with a maximum limit of 72 (25 °C and 50% relative humidity). When this limit is exceeded, milk production declines, since animals are under heat stress [104]. Heat has a bigger negative effect on cows than cold [67]. Despite this, cold stress can be also a common problem, especially in pasture-based farms or in farms with inappropriate facilities.

Cows experience cold stress with temperatures below −5 °C, with a reduction in milk production as a consequence [105]. Cold stress is primarily a major problem in calves, due to their natural inability to control their own body temperature [67]. According to [67,103,104,105,106], and represented in Figure 4, there is a representation of the temperatures limits, from which adult cows and calves start to suffer from thermal stress (heat and cold).

In an effort to solve thermal stress, indoor systems arose as a solution to the outdoor thermal stress. Nevertheless, indoor heat stress is still a major problem if farms do not implement appropriate methods to produce forced convention and evaporative cooling, such as fans and misting fans [107]. The creation of stables dates back to the beginning of the 20th century, using extremely laborious, individual barns [5]. From 1970 onwards, large feedlots were created, which were spread throughout the dairy world, with the advantage of allowing the handling of animals in groups, depending on their production stage [108]. All of this has driven a great evolution in the management and practices of the dairy sector [5,108].

On the other hand, indoor systems are linked to higher productivity, boosted by a better nutritional control, and improved reproductive management [97,99]. These systems also have the advantage of protection against extreme weather conditions [99] (Figure 3). Animals reared indoors were also less susceptible to gastrointestinal parasitism, such as *Fasciola hepatica* and gastrointestinal nematodes [93,104,109]. As example, a higher proportion of dairy herds with access to pasture presented strongyle (94% vs. 40%; *p* < 0.01), *Nematodirus* (82% vs. 20%; *p* < 0.01) and *Moniezia* (12% vs. 0%; *p* < 0.01) diagnosis, when compared with farms with indoor systems, exclusively [110]. However, in this study [110], *Trichuris* diagnosis was more frequent in confined farm systems (65% vs. 100%; *p* < 0.05).

Despite having improved several aspects, indoor systems have also shown varied constraints. Animal density must be taken into account when stabling the animals. Many farmers end up having too many animals per surface space to maximize the use of their facilities [111], which leads to great losses [112]. Animals in a high stocking density are more stressed and become more aggressive, especially due to less manger space and lying space disputes [113]. Farms with a lower or appropriate stocking density have animals with normal behavior, which lie down for longer periods, and show greater ruminative comportment [114,115]. In addition, the concentration of disease-causing micro-organisms or particles in the environment is inversely proportional to the space available to animals [116]. The incidence of pathologies such as lameness, mastitis, and uterine disorders also increases with stocking density [96].

Lameness is one of the three most prevalent disease in dairy farms, leading to large economic losses, with reduced yield and reproductive performance, and an important animal welfare constraint [117]. Cows with hoof diseases, especially in postpartum, have longer calving intervals, and a higher number of inseminations per conception [118]. Lameness is also related with lower body conditions scores [119] and poor indexes of animal welfare, which can be mitigated by preventive measures such as regular claw trimming [120]. A case of lameness can represent losses of between EUR 100–190 per case per year [121].

The increase in production diseases, such as acidosis and ketosis, and infections is another negative impact of intensive production systems, being a major problem in high-producing dairy cows [122]. They are mainly related with intense negative energy balance due to high yielding [123,124].

Table 1 summarizes the main advantages and disadvantages associated with each production system.

The best system varies from region to region, and several factors should be taken into account. Climate conditions, the availability and prices of land and raw feed materials, and production costs are all factors that influence the adoption of a system. Consumers’ perspective is a mean driver of market frames and, independently of which level of excellence we achieve with one system or another, according to the society, pasture-based farms provide animals with better conditions, enhancing health and welfare [125].

Therefore, grassland systems have an intrinsic worth in markets which are able to give value to this, as is the case for example in the Dutch market, where farms with grazing cows receive an increase in the final price of milk, compared to indoor farms [108].

### 3.5. Nutritional and Feeding Management

The nutritional and food management of animals has also undergone a huge evolution, as a result of new science developments and new knowledge [126]. These changes correlate with the availability and cost of raw materials. Moreover, the continuous raise in feed costs constitutes a threat to nutritional management and to farms’ profitability [127]. The feeding management verified in each farm, invariably depends on the mode of feeding, i.e., pasture-based or indoor cattle, and also depends on the size of the herd [128]. Dairy farms in poor landscapes, or where the lands have a high purchase or lease cost, can only buy external forages and food supplies [129,130].

Furthermore, when a nutritionally balanced diet is provided with less variability and higher precision, producers are able to extract the full potential of their animals [131]. The diet of these animals is highly controlled, and consists of energy rich foods, such as concentrated feed, corn silage, grass, and soy cuts, amongst others [132,133]. On the other hand, pasture-based farms have lower production costs, but with lower yields [98,131]. However, the need for the owners of these farms to increase productivity has led to the supplementation of forage (e.g., corn silage), and concentrated feed, thus increasing the dry matter (DM) intake [100,134].

The DM is the final mass of a food or product, after subtracting the water [135]. The evaluation of this parameter has been a very important tool for increasing the performance of animals. Through ingested DM, producers and nutritionists can calculate the required and actual intake, and how efficiently they are transforming it [136]. To increase and optimize DM intake, automatic feeding systems (AFS) were created [137]. These systems make it possible to reduce the farm’s management costs, reduce the workload, and above all, allow for an increase in the frequency at which the food is distributed by stall [138]. This increase in frequency allows the animals to distribute food intake more evenly, instead of having feeding spikes, with the advantage that the cows have a regular ruminal pH, throughout the day [139]. Lactating cows are usually ad libitum fed, in order to optimize DM intake (Figure 5).

According to Phillips et al. [140], increasing feeding frequency enhances the amount of milk produced, as well as the concentration of fat in milk; however, a similar study carried out in Finland [141] concluded that, although the animals boosted their efficiency in food conversion, there were no changes in the amount or concentration of the constituents of the milk produced.

On the other hand, Mattachini et al. [138] concluded that an excessive increase in food distribution can affect the resting time of animals, and also affect behavior. Farms with AMS, such as those already described, together with AFS, allow animals to show their natural behavior [139]. In fact, appropriate nutritional assessment and management are essential for animal welfare, productivity, and reducing the prevalence of pathologies, such as mastitis and reproductive problems [142,143].

### 3.6. The Milking Process

The historical evolution of the dairy sector is closely linked to the evolution of milking. The milking routine is one of the most laborious and precise processes that make up a dairy farm, corresponding to approximately 33% to 57% of the total work carried out daily on the farms [144]. Milking cows manually was hard work, and required a lot of labor, constituting an obstacle to farmers, who wanted to increase their herds and yield [5]. The necessity to raise productivity was driven by the need and demand for more products to feed the growing population at that time. All of these factors, but also the requirement for more secure and hygienic products, led to the appearance of the first mechanized milking systems at the beginning of the 20th century. Those systems were known as “milking catheters” [145].

Despite being a great advancement for that time, it caused distress [146], but also many problems of infections of the mammary gland, and the spread of mastitis due to bad milking practices, which at the time were little studied or known [145]. With the development of vacuum around the mid-20th century, great technological advances in milking systems were achieved, such as the “Rotolactor” in 1930 [5], and the Herringbone parlors in New Zealand [145].

From 1954 to 1964, several studies were conducted to know and understand not only the main causative factors of mastitis, but also to control bacterial infections in the udder. Such knowledge has allowed for an improvement in milk quality and an upgrade in udder health [147]. This brought enormous benefits to farmers, who were able to increase yield and milk quality. In the early 1990s, the first automatic milking systems (AMS), or milking robots, were presented in The Netherlands, being one of the greatest advances that the dairy industry had witnessed until then [11]. This responded to the need not only to optimize the time of farmers, but also to address the lack of labor for agriculture, which exists in many regions of the world [10,11].

The AMS offer greater flexibility in terms of the milking time, and allow for an increase in individual milking frequency, instead of the two or three milking/day normally carried out in the conventional system [148]. Since then, a lot of discussion between farmers and specialists in the area has arisen. Studies carried out indicate that the farms that have AMS register greater quantity and better bacterial quality of milk produced, as well as saving labor [148,149]. In turn, these systems imply higher repair and maintenance costs, as well as the need for specialized staff [150]. A recent study conducted on a farm in the United Kingdom found that after installing an AMS, there was a 13% increase in yield and a 28% reduction in somatic cell count (SCC) [151], with variations depending on farms. The enormous ability of the robot to provide varied data on the health and productivity of animals, is noteworthy [152].

The particular experience in the Azores is that many farms have mobile milking systems to milk in the pastures [8], and the implementation of AMS is not yet common. These mobile milkking systems are particularly and traditionally seen in smaller farms. Traditionally in the Azores, cows are milked in the pastures twice a day; the milk is transported in non-refrigerated milk tanks, and placed in the collection point of the local dairy industries (Figure 6) [8]. On the other hand, bigger Azorean farms are abandoning the pasture-based systems, opting for full indoor or semi-indoor production systems with milking parlors.

Regardless of the choice of milking systems, it is essential to have an adequate regulation of the pressure and vacuum systems [144], as well as the adoption of good milking practices and hygiene [153]. In line with this progress, improved milking practices have been introduced. Simple practices, such as wearing gloves, cleaning the udder, drying teats, use of post-dipping, and teat disinfection, among others [154], significantly control the spread of many pathogens inside farms [155]. As a result, a lower incidence of clinical mastitis, lower mammary gland infection rates, and decreased SCC are some of the benefits that come from adherence to these programs, which encompasses good milking practices, hygienic practices, and the strategic use of antibiotic therapy at dry-off [156].

## 4. The Control of Production Programs

Control of production programs is possible through a relationship of professionalism and cooperation between veterinarians and farmers [157], as well as other production-related professionals (e.g., nutritionists, production animal engineers). If, on the part of the farmers, there is up to a complete and organized record of all the daily occurrences and production indicators of the farms, on the part of the veterinarians, it is up to them to continuously update and analyze the general productive state of the farm [158]. This control can be optimized when using technologies capable of performing a detailed and accurate analysis of the data collected [159,160].

Currently, new technologies on the market are able to precisely manage livestock farms, thus ensuring effective control of production performance [160]. Precision livestock farming (PLF) is the latest production tool, designed to increase animal production [161,162]. The need to feed the rapidly growing world population arose the need to produce not only a rapid product, but also a safe and secure product [162]. The PFL systems ensure continued monitoring of farm production performance and animal health, being an efficient tool to warn farmers, when problems arise, reducing the time to take actions [161,163]. It is PFL’s main goals to increase farms productivity, based on animal welfare, health, and environmental sustainability [161].

Nevertheless, production programs are deeply interrelated with VHHM, and should be implemented and monitored as a whole.

## 5. Veterinary Herd Health Management and Future Perspectives

All of the previously discussed aspects led dairy farms to look for alternatives, and outline strategies to increase their yield and improve their efficiency. With this in mind, both producers and veterinarians have reformulated the management and their ways of acting. As such, on-farm VHHM programs were created and implemented all over the world [164].

The genesis of VHHM in day farms was an evolution of veterinary services and preventive veterinary medicine. The development of preventive veterinary medicine can be divided into four phases [165]: (1) the formalization of veterinary medicine in the late 19th and early 20th centuries mainly focusing on the eradication of clinical infectious diseases (e.g., brucellosis and tuberculosis); (2) the ambulatory clinics, from the 1940s, focused on individual medicine of production animals with the progressive availability of antibiotics. The vaccination of animals began to be used in the control of disease; (3) the recognition of subclinical forms of diseases as a major limiting factor of farms’ productivity, from the mid-1960s. Both veterinarians and farmers were more proactive, implementing of scheduled visits to implant infertility and mastitis control programs; (4) herd health programs were implemented and consolidated in farms from the late 1980s. These programs allowed to obtain an overall health status of the herd improving the animal welfare and rentability of the farms.

Over the last three decades, great advancements in science (e.g., nutrition, fertility, epidemiology, pathology), new technologic tools (e.g., automatization, precision livestock production and medicine, online database information and use), the structure of dairy industry (e.g., large-sized farms, production per dairy cow, genetic improvement), public health (e.g., food security), and demands from the market (e.g., trade of milk and dairy products, animal welfare, and less environmental impact) have overtaken the veterinary medicine approach, where the traceability of food is mandatory in most countries. The approach to clinical and subclinical disease during ambulatory clinics was greatly replaced by the maintenance of animal health using advice veterinary services, namely VHHM, to protect animals and people from illness, improve animal welfare, and reduce the environmental impact of the milk production [166], also mitigating antimicrobial use [167].

VHHM programs, which began in The Netherlands in the 1970s, basically consist of regular and previously arranged (and prepared) visits to farms, where a follow-up and consultancy service is provided for all areas intrinsic to milk production, reproduction, disease prevention, biosafety, nutrition, environment, and animal welfare [29,168], amongst others. With these visits, veterinarians are able to see the farm’s routine and, with a SWOT analysis approach, identify strengths and risk factors [169]. The ultimate goals of VHHM programs are to promote animal health, increase productivity/yield (with reduction in production costs), and disease prevention, with the recognizing and respecting of animal welfare, food safety, public health, and environmental sustainability [90,166,170].

VHHM programs are intrinsically related to the production plan of the dairy farms. LeBlanc et al. [166] characterized VHHM as “an integrated, holistic, proactive, databased, and economically framed approach to prevention of disease and enhancement of performance” of dairy farms, and also involving food safety and public health, animal welfare, and environmental sustainability. VHHM implantation and evolution, as well other veterinary services, has been primarily led by the dairy industry requirements throughout the last century, and is still rapidly changing to a new paradigm of One Health, as described in Table 2.

With the creation of the VHHM programs, there is a change in the classic role of the veterinarian, with less attention to the treatment or observation of a sick animal, and dedicating itself more to the general management of the herds [29,157]. With these programs, the priority becomes prevention, rather than treating a disease [166]. These programs have been able to adapt to the production system of each farm, i.e., whether indoor, pasture-based, or mixed dairy farms [29]. In fact, farms where VHHM programs are put into practice report higher production per cow, and better milk quality, with decreased SCC [168,171].

Some main points of the VHHM programs should be addressed at a dairy farm:(1)Milk quality programs: Mastitis is one of the most serious diseases in dairy cows worldwide [155]. This pathology is defined as an inflammation of the mammary gland, and can be caused by several bacterial and fungi agents, which lead to contagious mastitis or environmental mastitis [172]. We observe both clinical mastitis (with alteration in the consistency of the milk, in the udder, or in the animal) and subclinical mastitis (there is only an increase in SCC) [173]. Cows with this pathology represent significant economic losses (approximately EUR 120 per year), mainly due to decreased milk production, discarded milk, and medication costs [174]. Relevant advances in the diagnosis, treatment, and prevention of this disease have been achieved, such as improved management and increased hygiene in milking; correct management of dry and postpartum periods, and the development of vaccines against the main disease-causing agents [172]. Despite all these advances, mastitis remains one of the main causes of culling in dairy farms worldwide [175,176].(2)Peripartum health: The transition period, between the end of drying and the beginning of milk production, is a very sensitive moment in the dairy cow’s productive life. A greater energy supply is needed to meet the metabolic demands (for pregnancy and, later, for lactation), thus increasing the risk of metabolic diseases (e.g., hypocalcaemia and ketosis), and of infectious origin (e.g., metritis and mastitis) [177,178]. The low energy feeding in pre-partum period (poor dry cow management), and stress (social and nutritional), caused by the calving events, can lead to metabolic diseases as well [179]. According to Leblanc et al., 75% of dairy cows that fall sick, fall sick in the first month postpartum, an interval where a series of hormonal and metabolic changes occur, making them highly susceptible to sickness [166].(3)Biosecurity: It is crucial to prevent the introduction or spread of many multifactorial diseases within cows and within farms [180]. This control involves regular testing of the herd (serology, milk tank analyses, among others), quarantine and testing of purchased animals, hygiene and disinfection of spaces, and controlled visits to the farm, with the farm’s own clothing [181]. The implementation of biosecurity measures improves the health and welfare of the herds, with an increase in their productivity [182]. In addition, it is known that there is a reduced use of antibiotics on farms where these measures are applied [183]. This aspect is in line with the wishes of the Society and concretely with the European Commission, which aims to reduce the use of antibiotics and resistance to them [184]. The use of vaccines in the prevention of various pathologies can also be included in the batch of biosecurity measures. Vaccination plans were introduced in dairy farms in 1970, to reduce the incidence and prevalence of various pathologies. Dairy farms with effective and strict vaccination programs have higher productivity, and better fertility indicators [180,185,186,187]. It is in northern European countries that we see a greater adoption of vaccination and control protocols, with several national eradication programs successfully running. In fact, pathologies such as tuberculosis, infectious bovine rhinotracheitis, and bovine viral diarrhoea, amongst others, have already been eradicated in many regions [28].(4)Systematic reproduction control programs: These programs are associated with reproductive monitoring [188]. Reproduction is one of the essential pillars of dairy cows’ yield [177], since each calving is followed by lactation. Farms with poor reproductive performance indicators, incur large economic losses and high rates of culling [189]. The continuous reproductive monitoring conducted by veterinarians is mainly composed of routine pregnancy diagnoses, gynaecological evaluation (uterus and ovaries), mainly in the postpartum period, and monitoring of herd index, such as oestrus detection rate and pregnancy rate [190]. According to Inchaisri et al., inappropriate reproductive management, represents an estimated loss of EUR 231 per cow per year, which is due to the long calving interval, and the consequent decrease in milk yield [191].

A synopsis of the goals from each respective main point, covered by current VHHM programs, is reported in Figure 7.

To increase producers’ profits, the opportunity for veterinarians to provide holistic advice arises. Moreover, issues such as animal welfare, food safety, and antibiotic resistance, should be included in these plans [192]. Thus, the success and longevity of VHHM programs is the responsibility of these two professional groups. These programs have existed for over 30 years in central Europe, and are consciously adapted to the current requirements of the sector, and to evidence-based knowledge [168,171]. However, the relationship between producers and veterinarians is the key factor for its success [192].

Knowing the producer’s priorities, reliability, and trust are the three essential pillars for a good relationship and a successful partnership [157]. It is vital to understand what the producer’s objectives are, and their future perspectives. Unfamiliarity with farmers’ expectations and future objectives are the main factors of VHHM program failure [193]. Regular assessment is important, preferably by the same veterinarian, so that they can know and understand the main problems in each farm [174]. Veterinarians, in their approach to farmers, should be honest, direct, and objective [173,192]. The benefits, costs, and uncertainties of VHHM programs are now known, and they are a very important part of the work of veterinarians worldwide [194]. Moreover, veterinarians should base their work more and more on farm data management, personnel management, and their consulting and advisory skills should also be improved [195].

On the other hand, continuing education of farmers is needed. Farmers with greater knowledge have a greater ability to solve their own problems, and are able to take decisions more autonomously [157,166]. Besides that, these farmers better accept the advice given by veterinarians, and are more cooperative [192]. It is also important that farmers have organized and detailed recorded data of the occurrences of their farms. The processing of farm data, and the consideration of a sick animal as an indicator, may play an important role in the future for the early identification of disease [166].

As has already been said, in the present and future, topics such as animal welfare, reducing the use of antibiotics, environmental sustainability, and food safety, will inevitably become part of the daily life of dairy farms. Access to grazing, mass producing farms, or separation of the calf from the dam, are some of the current issues that are gaining a lot of consumer attention, along with an interest in the health and welfare of the animals. All the above-described evolutions, which led to a very specialized, mechanized, and intense management, caused a misalignment between the farmers’ practices, and what society considers as well-being [88]. It is up to the dairy sector to adapt to these changes, and give the consumer what they want, too. Farmers must be proactive, and must invest in new technologies, biosecurity, and preventive medicine, in order to provide and obtain high levels of health and productivity [28], without jeopardizing the medium-long term sustainability of their holdings. Farmers, veterinarians, and all those involved in the dairy sector must look at the new market needs not as threat or an obstacle, but as new opportunities to value themselves and their products [196].

Hazards and critical control points (HACCP) programs have also been implemented in dairy farms [197], such as in food industries [198]. HACCP programs establish the hazards, the risks, and the specific control measures, based on biosecurity and prevention [197,199]. This increases public health, food security, and also product traceability [200], but their implementation in dairy farms remains incipient mainly due to practical aspects and a large number of issues that should be covered. In recent years, the risk management system of specific issues has been reported, such as lameness [201], mastitis [202], paratuberculosis [203], and negative energy balance [204]. It is hoped that the focussed specific risk assessments based on the progressively evolving HAACP will be routine feature.

Finally, both veterinarians and farmers must incorporate their practices and actions, within the “One Health” perspective, living animal health, in accordance with human health and the environment [205]. Perhaps the pandemic situation we are currently experiencing reinforces this notion.

## 6. Conclusions

All of the changes and evolution that have been seen in the dairy sector have been implemented with the aim of making it more mechanized, specialized and highly productive. It is at the technological level that the biggest changes have been made in the sector, in particular in the milking process, and in food management. The creation of automatic milking systems (AMS) and automatic feeding systems (AFS) has made it possible to combat the lack of labor and abandonment of the sector, but also led to an increase in animal productivity. Its implementation should be encouraged, and its benefits better investigated. Furthermore, conditions for these systems to reach more areas of milk production should be created. In some areas, the lack of technical and expert support, makes it impossible for the implementation of these automatic systems.

The VHHM programs have and will continue to have a preponderance in the future of the dairy sector. They can be a very useful tool to farmers, in obtaining products certified in animal welfare and environmental sustainability, since there is a growing demand for these types of products. For this to be achieved, the management of herd health must be supported by prevention and biosecurity. Furthermore, animal health must be in symphony with environmental health and human health.

## Figures and Tables

**Figure 1 vetsci-09-00125-f001:**
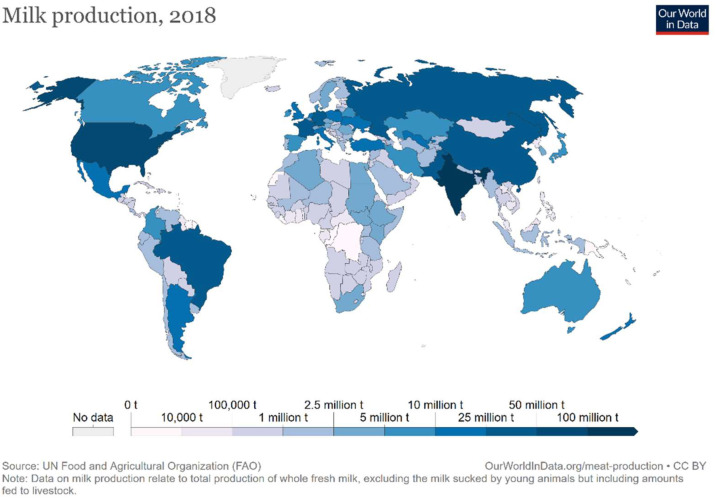
Milk production in the world. The interactive graph is free available at https://ourworldindata.org/grapher/milk-production-tonnes (accessed on 4 March 2022) and reports annual data of each country from 1961 [16].

**Figure 2 vetsci-09-00125-f002:**
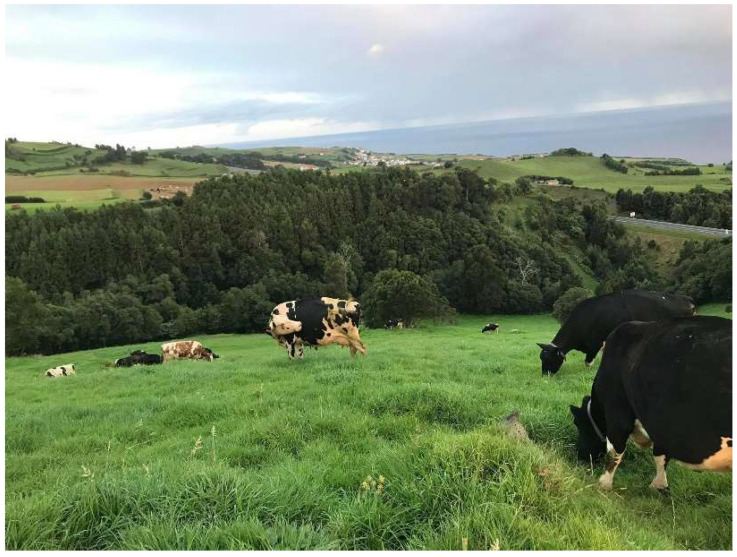
Grassing production system of dairy cattle (Azores).

**Figure 3 vetsci-09-00125-f003:**
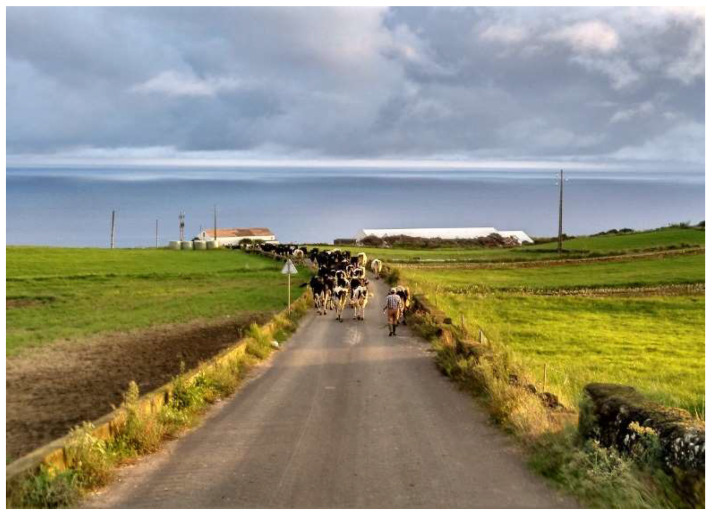
Transhumance of dairy cattle between non-contiguous grasslands (Azores).

**Figure 4 vetsci-09-00125-f004:**
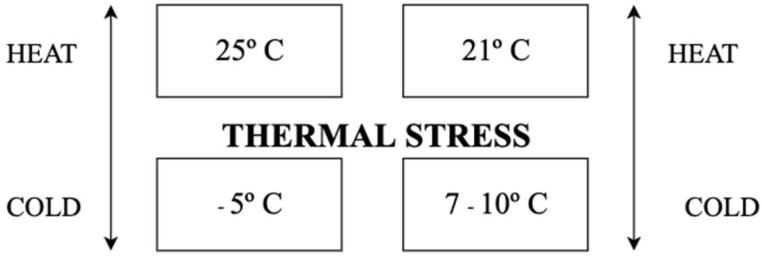
Thresholds of thermal stress in adult cows and calves.

**Figure 5 vetsci-09-00125-f005:**
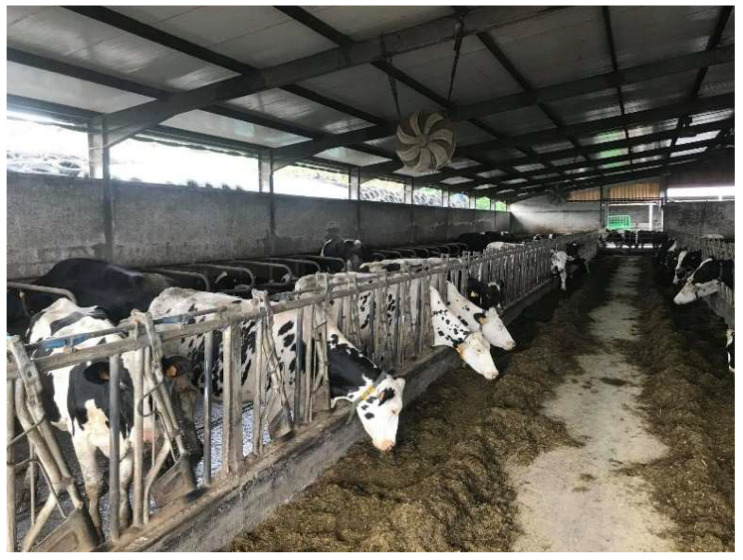
Feeding management using a total mixed ration system in confined in a low-sized farm (<200 lactating cows; Azores).

**Figure 6 vetsci-09-00125-f006:**
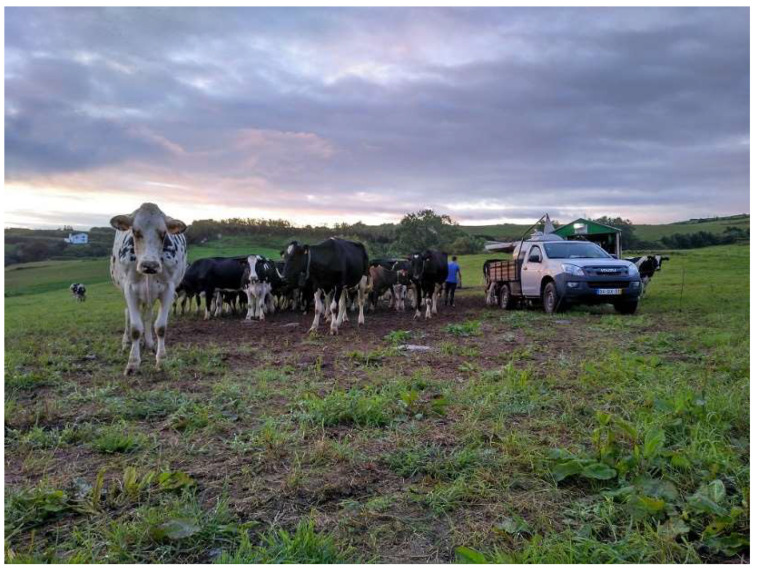
Mobile non-refrigerated bull tank milk (Azores).

**Figure 7 vetsci-09-00125-f007:**
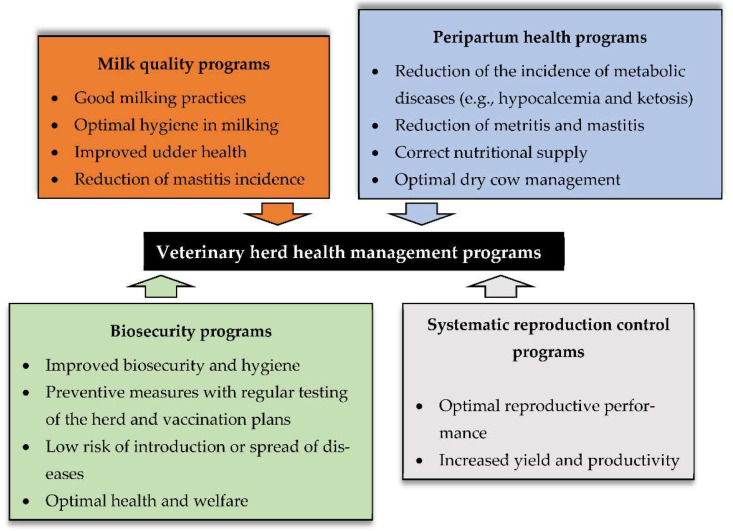
Goals of the four main aspects covered by veterinary herd health management programs.

**Table 1 vetsci-09-00125-t001:** List of main advantages and disadvantages related with pasture-based or indoor systems production.

**Pasture-based farms:**
Advantages: (a) Better animal health and welfare indicators (Crump et al. [93]). (b) Lower pathologies incidence, such as lameness (Arnott et al. [96]). (c) Low-cost production (Macdonald et al. [98]). (d) Consumer perspective: positive (Armbrecht et al. [125]). Disadvantages: (a) Limited protection against extreme weather conditions and predators (Charlton et al. [99]). (b) Higher propensity to thermal stress (Legrand et al. [104]). (c) Higher risk of gastrointestinal parasitosis (Crump et al. [93]). (d) Nutritional deficiency (Charlton et al. [99]; Crump et al. [93]). (e) Lower productivity (Charlton et al. [97]). (f) Weather dependence for grass growth (Wilkinson et al. [101]).
**Indoor farms:**
Advantages: (a) Higher productivity (Charlton et al. [97]). (b) Protection against extreme weather conditions (Charlton et al. [99]). (c) Lower propensity against thermal stress (Legrand et al. [104]). (d) Better nutritional control (Charlton et al. [99]). Disadvantages: (a) Higher incidence of production diseases such as lameness and uterine disorders (Arnott et al. [96]). (b) Occurrence of new production diseases, such as acidosis and ketosis (Steeneveld et al. [122]). (c) Stocking density problems (Charlton et al. [111]; Talebi et al. [113]). (d) Worse animal health and welfare (Crump et al. [93]). (e) Consumer perspective: negative (Armbrecht et al. [125]).

**Table 2 vetsci-09-00125-t002:** Timeline (surge or dissemination) of the main events influencing the evolution of preventive veterinary medicine and herd health management in bovine dairy production.

Year	Issue
	One Heath paradigm.
2010s–	Usage and availability of “big data” in the cloud.
	100,000 kg of milk per life production of dairy cow.
	Tentative implementation of hazard analysis and critical control point (HACCP) principle on dairy farms.
	Precision livestock production and medicine (e.g., sensors for estrous detection, in-line measurement systems for endocrine profiling).
	Antimicrobial use and resistance concerns.
	Mass dissemination of informatic tools for veterinarians and producers
2000s	New concepts in cow’s welfare and EU/national regulations
	Animal science and veterinary Medicine complementation in farm approach.
	Focus on the transition period.
	Epidemiological tools applied to dairy industry.
	Genetic improvement for relevant traits, including fertility (longevity and calving intervals) and health, other than milk yield.
	In vitro fertilization, multiple ovulation and embryo transfer.
	optimizing health and minimizing stress by nutrition improvements
	Continuing careful monitoring, appropriate biosecurity plans, appropriate and massive vaccination protocols.
1990s	Ovulation synchronization programs followed by fixed-time artificial insemination in herds (e.g., OVSYNCH protocols and it derivatives).
	Calf management and heifers’ replacement.
	Automatic milking systems (first commercial system in 1992, NL).
	Enlarged farms and progressive milk yield per standardized lactation.
	Reproductive tools using ultrasound scanning and hormone evaluation (milk progesterone tests, pregnancy-associated glycoproteins).
1980s	Use of somatic cell counts (SCC) and microbiology at udder and bulk tank milk levels, as indicators of intramammary infection.
	Herd health management implementations and dissemination (NL and US).
1970s	Conventional genetics programs to increase milk yield and animal conformation.
1950s	Computers as a management tool in dairy farming.
1906	First US milk recording association was founded.
1895	First report about records and collection of milk production data from a union of dairy farmers (DK).

## Data Availability

Not applicable.

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
