# Peer review of "Historical Evolution of Cattle Management and Herd Health of Dairy Farms in OECD Countries"

_vetsci, 2022, doi:10.3390/vetsci9030125_

Round 1

Reviewer 1 Report

I have carefully read the review: historical evolutión of cattle management and her health of dairy farms up to the present day. The authors emphasize  the improvement in the dairy industry and also point to the challenge to for farmers to obtain certified products with respect to health and welfare, as well environmental sustainability and future perspectives. I found it an interesting review and I think is an excellent article. The only aspect that they did not review and that would have been interesting is that of the contribution of genetics to improvement of dairy cattle, the control of production programs and the genetic evaluatión of breeders. 

Author Response

The genetic improvement as well as the genetic evaluation of breeders are described, now, from line 158 to line 185.

The control of production programs is summarized, now, from line 484 to line 503.

Reviewer 2 Report

The work has an ambitious goal as it deals with an extremely broad and complex topic, which would be better suited to a report or a book rather than a scientific article.

The authors' willingness to make a synthesis to be presented in a scientific article is commendable.

However, authors should review the entire article specifying well that their description relates to the European production system.
Since the development of the sector of the countries that are in emerging economies and the developing countries with climates different from those in Europe is very different.

The authors should decide whether the cut of the introduction which includes a world overview is well suited to what is an article which in detail describes the development of European (and non-world) breeds, genetic selections and productions.
My advice is to describe the production and consumption of the sector only that will be described later.

Small specific suggestions are indicated in the attached file

Author Response

Lines 2-3: The title was changed to “Historical evolution of cattle management and herd health of dairy farms in OCDE countries”.

In fact, a relevant part of this review addresses the EU production system. Nevertheless, the VHHM are related and influenced by other “developed” countries (e.g., see Table 2, and other requested alterations). So, we believe that the new title is more adequate to the content of the manuscript. 

Small specific suggestions are indicated in the attached file.

Line 32-45 In the “guidelines for authors” I did not find specified the possibility of including an Index. If they are required by the guidelines and I have not seen them, I apologize.

Otherwise. the index must be removed and a descriptive paragraph inserted in the introduction indicating what the presentation of the work will be.

Lines 66-70: The index was removed and the descriptive paragraph was added.

Line 47-49 It is necessary to specify where. Does the number of dairy cows mentioned and the relative production refer to the world, to Europe or to which territory? please specify

Line 33: “…worldwide…”

Line 35: “…in US…”

Line 115-119 The history of milk consumption (Although milk has been consumed historically….) should be moved to the beginning of the consumption paragraph starting on line 102

Moved to lines 92-96.

Line 120 2.2. The quota system described in paragraph 2.2 relates to the EU. it must be specified in the text and not just in the title

Line 114: “…EU implemented…”.

Reviewer 3 Report

Reviewer comments for manuscript ID vetsci-1591005 entitled ‘Historical evolution of cattle management and herd health of dairy farms’

General Comments

A nice review of the evolution of the dairy industry. Authors have tried to touch different aspects of cattle management and production. I congratulate the authors for making this effort.

An in-depth analysis of the evolution of the dairy industry is lacking. This review mentions some aspects without an analytical approach and herd health programmes required more thorough review. A timeline approach to the development of the dairy industry would have been better for comprehensibility and analysis.

I feel the review needs more work on the herd health programme evolution.

Specific Comments

Line 14: Please replace ‘main’ with ‘important’

Lines 19-20: Please reframe ‘All of this has been stimulated by the lack of labour, the need to increase farm’s efficiency and life quality of farmers’ as ‘Shortage of labour, emphasis on increasing farm efficiency and quality of life of the farmers’ were the driving factors for these changes’

Line 47: Please delete ‘responsible for’

Lines 48-49: Please reframe ‘when compared to the values produced 50 years ago’ with ‘in comparison to the 50-year-old production levels’

Line 52: Please delete ‘in this day and age’

Lines 54-55: Please reframe ‘with this being a solution to reduce fixed production costs’ with ‘reducing fixed production costs’

Line 56: Please rewrite ‘Several changes, and the introduction’ as ‘Several changes including the introduction’

Lines 60-61: Please reword ‘an abandonment of’ as ‘complete shift from’

Line 85: Please delete ‘one of the most competitive countries’

Line 86: Please replace ‘number of people’ with ‘strength’

Lines 92-93: Please replace ‘main manufacturer’ with ‘leading producer’

Lines 97-101: Are there other reasons for this during the COVID-19 outbreak, probably restricted outdoor movements and travel that might have increased consumption levels? Please clarify.

Lines 105-110: These are conflicting statements to the earlier assertions in the previous paragraph. Please clarify and rewrite.

Line 146: Please rewrite ‘worldwide breeds with dairy aptitude’ as ‘dairy breeds worldwide’

Lines 147-48: Please clarify what is meant by 2 more kg/cow of total protein and fat with 9 more kg/cow of total fat. I am sorry I am not able to understand the units.

Line 153: Please rewrite ‘its incredible ability for milk yield’ as ‘these traits’

Line 164: Please rewrite ‘has also been linked partially’ with ‘has lead to’

Line 174-74: Please clarify this ‘and have double aptitude’. Rather it should be ‘have dual purpose’

Line 175: Please replace ‘becoming an’ with ‘providing an’

Line197: Please rewrite ‘a significant part of these animals is’ as ‘these animals’

Lines 200-01: Please complete the sentence.

Lines 218-19: Please delete ’depending on studies’

Line 223: Please delete ‘gastrointestinal’

Line 321: Please replace ‘Nevertheless’ with ‘However’

Lines 35-53: Please reframe into simpler and smaller sentences.

Author Response

General Comments

A nice review of the evolution of the dairy industry. Authors have tried to touch different aspects of cattle management and production. I congratulate the authors for making this effort.

An in-depth analysis of the evolution of the dairy industry is lacking. This review mentions some aspects without an analytical approach and herd health programmes required more thorough review. A timeline approach to the development of the dairy industry would have been better for comprehensibility and analysis.

I feel the review needs more work on the herd health programme evolution.

Lines 511-538, 549-555: new information about evolution of the dairy industry and VHHM programs was added, including the Timeline approach in Table 2.

Lines 688-697: Also, a paragraph addressing HACCP programs in dairy cows was added to complete the overview.

Specific Comments

Line 14: Please replace ‘main’ with ‘important’

Line 14: Done.

Lines 19-20: Please reframe ‘All of this has been stimulated by the lack of labour, the need to increase farm’s efficiency and life quality of farmers’ as ‘Shortage of labour, emphasis on increasing farm efficiency and quality of life of the farmers’ were the driving factors for these changes’

Lines 19-20: Changed.

Line 47: Please delete ‘responsible for’

Deleted.

Lines 48-49: Please reframe ‘when compared to the values produced 50 years ago’ with ‘in comparison to the 50-year-old production levels’

Lines 34-35: Changed.

Line 52: Please delete ‘in this day and age’

Deleted.

Lines 54-55: Please reframe ‘with this being a solution to reduce fixed production costs’ with ‘reducing fixed production costs’

Line 40: Changed.

Line 56: Please rewrite ‘Several changes, and the introduction’ as ‘Several changes including the introduction’

Line 42: Changed.

Lines 60-61: Please reword ‘an abandonment of’ as ‘complete shift from’

Line 47: Changed.

Line 85: Please delete ‘one of the most competitive countries’

Deleted.

Line 86: Please replace ‘number of people’ with ‘strength’

Line 75: Modified.

Lines 92-93: Please replace ‘main manufacturer’ with ‘leading producer’

Lines 81-82: Changed.

Lines 97-101: Are there other reasons for this during the COVID-19 outbreak, probably restricted outdoor movements and travel that might have increased consumption levels? Please clarify.

Lines 88-91: All the causes to increase international dairy trade in 2020 were now reported for China according to [1].

Other causes from other counties are secondary and circumstantial (e.g., drought in Brazil). In general, the restriction of movement had a negative effect on milk trade.

Lines 105-110: These are conflicting statements to the earlier assertions in the previous paragraph. Please clarify and rewrite.

Line 102: “…, i.e., dairy products are not traditionally included in diet, …”. Cultural diets and lactose intolerance is relevant in this populational groups. Nevertheless, an increase in milk/milk product consummation increased during last decade such as reported in lines 104-108.

Line 146: Please rewrite ‘worldwide breeds with dairy aptitude’ as ‘dairy breeds worldwide’

Line 139: Changed.

Lines 147-48: Please clarify what is meant by 2 more kg/cow of total protein and fat with 9 more kg/cow of total fat. I am sorry I am not able to understand the units.

Lines 137-146. The units were g/Kg, but we changed the reference for one more appropriate (in %).

Line 153: Please rewrite ‘its incredible ability for milk yield’ as ‘these traits’

Line 147: Done.

Line 164: Please rewrite ‘has also been linked partially’ with ‘has lead to’

L171: Changed.

Line 174-74: Please clarify this ‘and have double aptitude’. Rather it should be ‘have dual purpose’

Line 195: We thank the reviewer because of this, and every previous punctuality. We changed it.

Line 175: Please replace ‘becoming an’ with ‘providing an’

Line 195: Changed.

Line197: Please rewrite ‘a significant part of these animals is’ as ‘these animals’

Line 224: Changed.

Lines 200-01: Please.

Line 227: “…makes it control very hard…”.

Lines 218-19: Please delete ’depending on studies’

Deleted.

Line 223: Please delete ‘gastrointestinal’

Deleted.

Line 321: Please replace ‘Nevertheless’ with ‘However’

Changed.

Lines 35-53: Please reframe into simpler and smaller sentences.

The index was deleted.

Round 2

Reviewer 2 Report

The authors made a great effort in trying to answer all the reviewers' requests.
The work is much improved.
As far as the work is concerned it can be published in the present form

Author Response

Reviewer 2

The authors made a great effort in trying to answer all the reviewers' requests.

The work is much improved.

As far as the work is concerned it can be published in the present form

We thank the reviewer for his/her comment.

Reviewer 3 Report

Reviewer comments for manuscript ID vetsci-1591005 entitled ‘Historical evolution of cattle management and herd health of dairy farms in OCDE countries’ - Round 2

General comments

I congratulate the authors for their hard work in revising this manuscript. It is much improved and provide a much better overview now. I am impressed with the timeline table. Nice effort. The changes /corrections suggested have been done by the authors. There are minor corrections that I have specifically mentioned. I recommend the publication of the manuscript following these corrections.

Specific comments

Line 68: I think it should be ‘EU where’

Lines 159-60: Please reword ‘allowed farms to massify’ as ‘enabled farms to improve’

Line 161: Please reword ‘greater productive capacities’ as ‘higher production traits’

Lines 168-69: Please reframe ‘When this cross resulted in genetically superior offspring, we were facing a genetic improvement’ as ‘This cross breeding resulted in genetic improvement through the production of in genetically superior offspring’

Lines 182-85: Please reframe ‘Not also it is a great tool for the selection of the desired traits and breed conservation, it is also used these days, as a solution pf the inbreeding index, which negatively impacts reproduction and production traits [47]’ as ‘In addition to being a better tool for the selection of the desired traits and breed conservation, it restricts inbreeding, which negatively impacts reproduction and production traits [47]’

Line 503: Please delete ‘as following described’

Lines 688-89: Please reframe’ had also been tried to implement’ as ‘have also been implemented’

Lines 695-97: Please reword ‘It seems that the evolution of HAACP will be done progressively by the sum of these focused specific risk assessments in the coming years’ as ‘It is hoped that the focussed specific risk assessments based on the progressively evolving HAACP will be routine feature’

Author Response

Reviewer 3

General comments

I congratulate the authors for their hard work in revising this manuscript. It is much improved and provide a much better overview now. I am impressed with the timeline table. Nice effort. The changes /corrections suggested have been done by the authors. There are minor corrections that I have specifically mentioned. I recommend the publication of the manuscript following these corrections.

Thanks to the reviewer. We also believe that the Table 2 shows the most relevant aspects of VHHM /preventive veterinary medicine throughout more than 100 years.

Specific comments

Line 68: I think it should be ‘EU where’

Corrected.

Lines 159-60: Please reword ‘allowed farms to massify’ as ‘enabled farms to improve’

Reworded.

Line 161: Please reword ‘greater productive capacities’ as ‘higher production traits’

Reworded.

Lines 168-69: Please reframe ‘When this cross resulted in genetically superior offspring, we were facing a genetic improvement’ as ‘This cross breeding resulted in genetic improvement through the production of in genetically superior offspring’

Changed.

Lines 182-85: Please reframe ‘Not also it is a great tool for the selection of the desired traits and breed conservation, it is also used these days, as a solution pf the inbreeding index, which negatively impacts reproduction and production traits [47]’ as ‘In addition to being a better tool for the selection of the desired traits and breed conservation, it restricts inbreeding, which negatively impacts reproduction and production traits [47]’

Changed.

Line 503: Please delete ‘as following described’

Deleted.

Lines 688-89: Please reframe’ had also been tried to implement’ as ‘have also been implemented’

Changed.

Lines 695-97: Please reword ‘It seems that the evolution of HAACP will be done progressively by the sum of these focused specific risk assessments in the coming years’ as ‘It is hoped that the focussed specific risk assessments based on the progressively evolving HAACP will be routine feature’

Changed.
